# Vascular Plant Species Inventory of Mexico’s Revillagigedo National Park: Awareness of Alien Invaders as a *Sine Qua Non* Prerequisite for Island Conservation

**DOI:** 10.3390/plants12193455

**Published:** 2023-09-30

**Authors:** Alejandra Domínguez-Meneses, Juan Esteban Martínez-Gómez, Teresa Mejía-Saulés, Israel Acosta-Rosado, Stefan Stadler

**Affiliations:** 1Red de Interacciones Multitróficas, Instituto de Ecología A.C. (INECOL), Xalapa 91073, Veracruz, Mexico; alejandra.dominguez@posgrado.ecologia.edu.mx; 2Red de Biología Evolutiva, Instituto de Ecología A.C. (INECOL), Xalapa 91073, Veracruz, Mexico; teresa.mejia@inecol.mx; 3XAL Herbarium, Instituto de Ecología A.C. (INECOL), Xalapa 91073, Veracruz, Mexico; israel.acosta@inecol.mx; 4Frankfurt Zoo, 60316 Frankfurt am Main, Germany; stefan.stadler@stadt-frankfurt.de

**Keywords:** Eastern Pacific, exotic plants, invasive species, invasiveness risk, oceanic islands

## Abstract

The Revillagigedo Archipelago, located in the Eastern Pacific Ocean, stands out for its unique biological richness and endemism. These islands remained uninhabited until the second half of the twentieth century, allowing a better conservation status than on other oceanic islands. However, the continuous introduction of potentially invasive alien plant species, and the lack of adequate control or eradication actions, jeopardize the conservation and restoration of these islands’ fragile ecosystems. We present the most complete vascular plant species inventory and an updated list of alien plant species of the Revillagigedo Archipelago, which was compiled through an extensive review of national and international plant collections and other sources. Our 272 species list includes 106 alien plant species (39.3%; 104 in Socorro, and 16 in Clarion): 67 (24.8%) are naturalized, 14 (5.2%) are casual aliens, and 25 (9.3%) subsist under cultivation. The documented alien species belong to 73 families. Annual and perennial herbs are the prevailing life forms in the alien flora, while naturalized species are primarily native to North America. The number of introduced species has increased significantly since the islands became inhabited. Many of the recently introduced species pose a major invasion risk like on other islands of the world.

## 1. Introduction

The Revillagigedo Archipelago, located in the eastern North Pacific, is a group of four oceanic islands: Socorro, Clarion, San Benedicto, and Roca Partida [1]. These islands are famous for their invaluable biological richness, which, thanks to their isolation from the continent, led to the evolution of a high number of endemic species [2], as well as many protected and endangered species [3]. Declared as a Protected Natural Area in the category of Biosphere Reserve in 1994 [4], the Revillagigedo Archipelago entered the World Heritage List of the United Nations Educational, Scientific, and Cultural Organization in November 2016 [5]. In November 2017, it was declared a National Park [6], and with a surface of 148,087 km^2^, it is the largest marine sanctuary in North America free of fishing activities [7].

Herbivores were introduced to these islands in different periods. Domestic sheep (*Ovis aries*) were brought to Socorro in 1869 [8]. Overgrazing by sheep facilitated the transformation into open and eroded habitats of a significant percentage of the forests and scrublands in the southeast half of the island [9]. The drastic impact on the island´s vegetation probably played a major role in the extinction of the Socorro Elf Owl (*Micrathene whitneyi graysoni*) in the 1930s [10] and the extinction of the Socorro Dove (*Zenaida graysoni*) in the wild in the mid-1970s [9]. After the removal of sheep in 2012, passive recovery of the vegetation cover has been observed in the most affected areas in the south of the island. However, a considerable amount of this recovery does not involve native species [11].

Clarion was also subjected to herbivore introductions, domestic pigs (*Sus scrofa*) in 1980, domestic rabbits (*Oryctolagus cuniculus)* in 1983, and sheep taken from Socorro between 1990 and 1991 [12]. The introduction of these mammals affected the island’s vegetation drastically, and some native plant species, like *Opuntia* sp., have diminished drastically [13]. Although pigs and sheep have been eradicated [14], the transformed landscape is more susceptible to the establishment of alien plant species.

As on other islands, many alien plant species were most likely introduced unintentionally, with their seeds traveling in the animals’ fodder, wool, fur, and excrement, as well as crop contaminants [15]. The absence of permanent human settlements, until the establishment of a naval garrison on Socorro in 1957 [1] and a naval station on Clarion in 1979 [12], prevented the intentional introduction of alien plants. Afterwards, more alien plant species arrived due to human intervention, even though introductions were strictly prohibited since the publication of the management program of the Biosphere Reserve in 2007 [16] and, more recently, with the establishment of the National Park [17].

Several botanical expeditions have been conducted in the Revillagigedo Archipelago since the XIX century [18]. Since then, relevant work has been carried out to compile and describe the archipelago’s flora, for instance, Johnston [19] and references therein, Levin and Moran [1], and Rebman et al. [20]. Additionally, other valuable contributions have been made, such as the revision of pteridophyte species by Mickel and Smith [21], or punctual additions by Dávila et al. [22] and Benavides et al. [23]. Despite these studies, the Revillagigedo Archipelago still lacks a comprehensive plant inventory that integrates all of the confirmed available information and an up-to-date list of alien plant species.

Island ecosystems are particularly vulnerable to invasions by alien plant species; these species bring negative impacts to the native flora and, subsequently, to the native fauna, particularly specialist and endemic species [24]. The Revillagigedo Archipelago hosts several exotic plant species known to be extremely invasive in other islands. For instance, the guava tree (*Psidium guajava*) is one of the most harmful invasive species in the Galapagos Archipelago [25]; however, its potential impacts on Socorro remain unassessed.

The successful management of invasive plant species requires active work to prevent new introductions, such as surveillance to detect emerging populations and persistent efforts to eradicate or at least control the worst invaders [26]. Even though it is understood that invasions are context-specific, and invasiveness will only materialize when certain environmental conditions are met [27], invasive patterns of species elsewhere can help anticipate how they will behave when introduced into a new territory [26].

In this article, we present the most complete and up-to-date list of native and introduced flora of the Revillagigedo Archipelago. Furthermore, this study aims to categorize alien plant species through their known invasiveness risk to forecast their impact on the native vegetation in the archipelago and to prevent possible negative outcomes. We studied and compared the invasion patterns on Socorro and Clarion across time to propose a way to establish management priorities while taking notice of limited logistical and financial resources. Also, the status of native and alien plant species of the Revillagigedo Archipelago is compared with other insular regions of the world.

## 2. Materials and Methods

### 2.1. Study Area

The Revillagigedo Archipelago is composed of four volcanic islands: Socorro, Clarion, San Benedicto, and Roca Partida (Figure 1). Socorro, the largest and most diverse island, is located 700 km west of Manzanillo, Colima [28]. It has an area of 131.32 km^2^ [29]. The climate from 0 to 400 m is tropical semi-arid, while from 400 to 1150 m, it has a sub-humid subtropical climate [30]. The annual average temperature is 24.7 °C. The average annual accumulated precipitation is 313.8 mm; most precipitation occurs between July and October and the dry season is observed between February and June. During the rainy season, tropical storms and hurricanes are frequent [28].

The vegetation of Socorro has been classified into nine types [28]: coastal halophiles, *Conocarpus erectus* scrub, grasslands (*Cenchrus ciliaris*, *Cynodon dactylon*, *Melinis repens* [31]), *Croton masonii* scrub, *Pteridium caudatum–Dodonaea viscosa* scrub, mountain mesophilic forest (*Ilex socorroensis* and *Guettarda insularis*), deciduous tropical forest (*Ficus cotinifolia*), evergreen tropical forest (*Psidium-Sideroxylon* and *Hippomane mancinella*), and prairie (*Coreocarpus insularis*, *Lepechinia hastata socorrensis* and *Brickellia peninsularis*) [28]. 

Socorro’s fauna consists of seabirds, migrating shorebirds, seasonal transients, eight endemic terrestrial birds [32], and one reptile, the endemic Socorro Lizard (*Urosaurus auriculatus*), as well as numerous species of invertebrates, among which the land crab (*Johngarthia planata*) is the most representative. The exotic fauna consists of two bird species, the Mourning Dove (*Zenaida macroura*) and the Northern Mockingbird (*Mimus polyglottos*), two mammals, namely, feral cats (*Felis catus*) and the domestic mouse (*Mus musculus*), and one reptile: the Tropical House Gecko (*Hemidactylus frenatus*).

The human population on Socorro is comprised of personnel from the Mexican Navy (between 40 and 70 people), who live in the naval garrison in the island´s southeast; additionally, there is an intermittent presence of civilian personnel and researchers. Food consumed by this population is transported from the continent, with very few exceptions, like fruits of introduced species (guava, coconut) and some species of fish. In the past, drinking water was transported to the island by ship; today, it is obtained from a desalination plant located at the naval garrison (Martínez-Gómez, unpublished field notes).

Clarion is the furthest island from the mainland and second in both size (19.58 km^2^) [29] and biodiversity [12]. The highest point on Clarion reaches 308 m, while the elevation on the rest of the island averages around 150 m. The vegetation on this island is primarily herbaceous (*Brickellia peninsularis*), with only a few shrubs like *Rhamnus humboldtiana* with a height of no more than 3 m [12]. Its fauna includes eight bird species, which breed on the island, four of them being Clarion endemics, and several bird migrants [14], the endemic lizard (*U. clarionensis*), and two endemic snakes (Clarion Island Racer, *Masticophis anthonyi* and the Clarion Nightsnake, *Hypsiglena unaocularus*) [33] as well as insects, spiders, land snails, land crabs and introduced European rabbits [13], and the Western Spiny-tailed Iguana (*Ctenosaura pectinata*) [34]. Clarion is inhabited primarily by Mexican Navy personnel, whose number fluctuates between 10 and 20 people, and as on Socorro, their food is transported by ship (Martínez-Gómez, unpublished field notes).

The remaining two islands lack introduced plants. San Benedicto is a small island of 6.09 km^2^ [29], with a maximum altitude of 335 m [35]. One of the volcanoes of this island, the Barcena, erupted between 1952 and 1953, destroying most of its original vegetation [35]. Since this event, the vegetation has recovered [35]. Unlike Socorro and Clarion, it does not seem to have been colonized by alien plant species [20]. San Benedicto hosts large numbers of seabirds, occasional bird migrants, and visitors from Socorro, as well as land crabs, insects, and spiders [13]. It does not have a permanent human population and it is not visited as frequently as Socorro or Clarion. Roca Partida is an islet of only 0.02 km^2^ [29] with a maximum height of 50 m. It lacks vascular vegetation and its fauna consists only of resting seabirds [13].

### 2.2. Inventory of Plant Species Present on the Revillagigedo Archipelago

We built an inventory of the vascular plant species of the Revillagigedo Archipelago, compiled from published lists [1,19,20,21,22,23,36,37]. Each published species was corroborated with herbarium specimens or photographic records. We collected plant specimens during expeditions conducted on the archipelago by members of our team over an extended period. Initially, we visited Socorro during July 1993 and March–May 1994, and visited Socorro and Clarion during March–April 1995. After a pause, we resumed our collecting efforts on Socorro during December 2013–January 2014, December 2017–January 2018 [31], and November–December 2018. We visited San Benedicto in December 2014–Jan 2015 and January 2016 to obtain photographic records of the vegetation. Then, we carried out a comprehensive review of herbarium specimens from the archipelago in the Enciclovida.mx website [38] of the National Information System on Biodiversity (SNIB), and consulted multiple herbaria from Mexico and the United States (Appendix A). Current names, basonyms, or synonyms were validated with the following global floristic databases: The Plant List website [39], World Flora Online website [40], the International Plant Names Index website [41], and the Plants of the World Online website [42].

The introduced plant species compiled in our inventory for the Revillagigedo Archipelago were assigned a category adapted from Richardson et al. [43] and Pyšek et al. [44] to designate their invasion status, whereas *alien* species are those whose presence is assumed to be the result of human-mediated transport. We decided to include those alien species that were introduced for horticulture, forestry, or ornamental purposes and have not escaped from cultivation, under the category of *Cultivated* [45], so we lay a precedent over the presence of potentially invasive species, although for now, they are only represented by a few individuals living under cultivation. *Casual* alien species are alien plants that may flourish and even reproduce occasionally outside cultivation, but do not form self-sustaining populations and may rely on repeated introductions for their persistence.

Taking into consideration the isolation of the archipelago, we reduced the minimum residency time required to consider a plant as *Naturalized*. In this category, we included species that have self-replacing populations for at least five years, as in the case of recognized highly invasive species with already large numbers or formed groups at considerable distance from their parental plants. Also, they have the potential to spread over a larger area, without direct human intervention, by recruitment from seeds or ramets capable of independent growth [27]. When a species was present on more than one island with different invasion stages, the highest status was applied [46]. Although invasive plant species are certainly present on the Revillagigedo Archipelago, we did not assign the category *Invasive*, because there is not enough information available on the spread rates and impacts of naturalized species for this region. Instead, we performed an invasiveness risk assessment based on the available information for each species from other insular regions (see below).

### 2.3. Species Traits: Origin, Life History, and Date of Introduction

The putative origin of alien plants in this inventory, as well as their life form and habit, was determined using the following global floristic databases: World Flora Online [40], Plants of the World Online [42], and the International Plant Name Index [41]. The native range of the species was grouped into seven regions resulting from the classification of major biogeographically defined areas of the Biodiversity Information Standards Organization (originally Taxonomic Databases Working Group [47]): North America (including Mexico), South America (including Central America), Africa, Europe, Tempered Asia, Tropical Asia, and Australasia. Since many species are native to more than one region, they were assigned to all regions where they are considered native [46]. Plants whose native range is unknown because they are a result of recent human-mediated hybridization were not included in any category. Since cultivated species are not yet found in the wild, they were excluded from the analyses of habit, life form, and date of introduction [48], but were considered in the origin determination and invasiveness risk assessment. Habit was classified as *Grass*, *Herb*, *Vine*, *Shrub*, or *Tree* and life form as *Annual* (for annuals and biennials) or *Perennial* [49].

In order to define an approximate date of introduction, we searched for the year of the first published record, or the first collected specimen available at herbaria [45], both for native and introduced species. Then, we built native and introduced species accumulation curves across years, for both Socorro and Clarion, and compared the distributions of these curves with Kolmogorov–Smirnov two-sample tests [50,51] using the R “fBasics” package [52]. These tests, which provide the cumulative probability that two samples have the same distributions [53], were used to assess whether native and introduced species have been registered in a similar way over time, as the number of scientific expeditions increased, or if introduced species records respond to other factors, such as the establishment of human settlements on the islands or species introduction restrictions. The proportions of native and introduced species in 30-year periods for both islands were estimated from the total number of species known for each period. A multiple sample test for equality of proportions was performed using the R “stats” package version 4.0.3 [54]. We ran *post hoc* tests to identify which proportions were different using the routine binom_test with a Holm–Bonferroni adjustment for multiple comparisons, with an experiment-wise significance level set at α = 0.05.

### 2.4. Invasiveness Risk Assessment for Alien Plant Species

Weed Risk Assessment (WRA) systems, originally created to evaluate the invasiveness risk of plant species before their introduction into certain island countries [55,56,57], can also be used to estimate the invasiveness risk of recently introduced alien species in a new environment. An example of this approach is the categorization of plants for their invasiveness risk as presented in the Global Compendium of Weeds [58]. This compendium considers a list of all invasive plant species anywhere in the world, along with their characteristics, such as the means of entry of plants into new environments, the routes of dispersal and the impact they have had on ecosystems, and the relationship the plant has had with humans [59]. This scoring system calculates a plant’s risk potential as a ranking that can be compared to other weed species [58].

In addition to classifying alien plants by their invasion status in the archipelago, we included an invasiveness risk assessment for every species using the rating proposed in the latest edition of the Global Compendium of Weeds [58]. This rating is obtained by calculating a score, which grades each species in terms of (1) its ability to enter new environments, (2) to spread and establish over wider areas by its dispersal options, and (3) by its impacts in these environments. This score allows one to rate invasiveness risk as *Extreme*, *High*, *Medium*, and *Low*. Since this system owes its strength to the number of times that plant has been historically reported as a weed in different regions of the world, there is a possibility that some species are included in the compendium, but are not rated because there are not enough records to assign a score.

### 2.5. Comparison of the Status of Native and Alien Plant Species of Different Insular Regions

To compare trends among different insular regions of the world, we evaluated the relationship between the number of species versus island area for both native and alien plants. We fitted a generalized linear model (GLM) using the quasi-Poisson family with a log link function [60]; the Poisson regression is appropriate when modeling an overdispersed count variable. The number of species was used as the response variable and island area as the independent variable. Analyses were performed separately for native and invasive species. Island size was obtained from INEGI [29] and other sources [61,62,63,64,65,66,67,68,69].

## 3. Results

### 3.1. Inventory Overview

Our work allowed us to assemble the most accurate list of vascular plant species for the Revillagigedo Archipelago. We eliminated spelling and taxonomic errors from existing lists, like double or even triple records of the same species that were, in fact, synonyms or basonyms, as well as non-existent species. Our final compilation is based on a thorough revision of published listings and herbarium specimens and resulted in 272 species for the entire archipelago belonging to 73 families (the complete and substantiated species list, by island, is available in Appendix A). Socorro has 237 species, Clarion has 62, and San Benedicto has 11 (of which four endemic and one native species are believed to be absent from the island since the eruption of Barcena Volcano in 1952) [35]. Thirty-three species are strictly endemic to Socorro, eight to Clarion, three to Socorro and Clarion, four to San Benedicto, and four to the Revillagigedo Archipelago. We found 106 alien species, four of which were collected on Socorro, but have not been observed since 2010 (Martínez-Gómez, unpublished field notes; Table 1). Six species were not included in our inventory because they have only been mentioned in the species catalog compiled by Captain Ignacio Cabrera [16]; these species lack herbarium or photographic support and other reports did not confirm their presence. Although some authors suggest treating insular forms and island varieties as a single species or as equivalents of their continental counterparts (e.g., González-Gallegos et al. [70] and Wood et al. [71], we consider them distinct forms when noticeable morphological differences have been observed previously (sensu Levin and Moran [1]).

The 106 introduced species included in our inventory represent 39% of the total species present on the Revillagigedo Archipelago. When considering only naturalized species, they account for 29% of the total species present (0.41 naturalized/native ratio). Regarding Socorro, naturalized species represent 28% of the total species richness (0.5 naturalized/native ratio), while for Clarion, they represent 21% (0.28 naturalized/native ratio).

Socorro accounted for 104 alien species belonging to 39 families, among which the most represented families are Fabaceae (18 species), Poaceae (14 species), Malvaceae (13 species), and Euphorbiaceae (five species). In total, 66 alien species were classified as naturalized, 14 as casuals, and 24 as cultivated. Clarion presented 16 alien species, belonging to nine families, among which the most represented families are Poaceae (six species) and Fabaceae (three species) (Figure 2). Thirteen species were classified as naturalized and three as cultivated. San Benedicto did not have any alien species.

### 3.2. Origin and Life History

The majority of alien species in the Revillagigedo Archipelago have their native range in the Americas (Table 2). North America contributes 66 species and South America 58 species. Africa and Asia follow; Africa contributed 32 species, Temperate Asia 32 species, and Tropical Asia 27 species. Australasia with 13 species and Europe with 5 species were the least represented regions. North America, Africa, and Asia were the regions with most naturalized species.

The most common life history among the naturalized and casual species of Socorro are annual herbs (24 species; 30.0%) and perennial herbs (18 species, 22.4%; Figure 3). Perennial woody species account for ten trees (12.5%) and seven shrubs (8.7%), while there is only one annual shrub (1.3%). Grasses are represented by nine annual species (11.3%) and six perennial species (7.5%). Vines are represented by four perennial species (5%) and one annual species (1.3%). On Clarion, grasses are the most common naturalized species, with four annual (30.7%) and two perennial species (15.4%). Herbs account for three annual species (23.1%) and two perennial herbs (15.4%), while vines are represented by two perennial species (15.4%).

### 3.3. Year of Introduction

The dates of first collection or first report available for each species were used a proxy of the year of introduction. On Socorro, only 11 of the 80 casual and naturalized species were reported before the naval garrison was established in 1957. Fifty-three species were first reported between the establishment of the naval garrison and the publication of the Biosphere Reserve management program in 2007 [16], which prohibited the introduction of exotic species. Twelve more species have been reported since (Figure 4a). On Clarion, two introduced species were reported before the establishment of the naval garrison in 1979, another five species were reported between 1979 and the publication of the management program in 2007, and six species have been reported since its publication (Figure 4b).

Kolmogorov–Smirnov tests showed that the species accumulation curves of native and introduced species on Socorro and Clarion belong to statistically different distributions; native and introduced species have not been recorded in a similar way over time (D = 0.7293, *p* ≤ 0.001 for Socorro, D = 0.8579, *p* ≤ 0.001 for Clarion; Figure 4 c,d). The test for equality of proportions yielded significant differences between the introduced/native species over five 30–year periods on Socorro (χ^2^ = 61.41, df = 4, *p* ≤ 0.001; Figure 5). The Holm–Bonferroni-adjusted *p*-values yielded a significant difference between the periods spanning from 1881 to 1970 and from 1971 to 2021 (*p* ≤ 0.001; Figure 6). For Clarion, although the test for equality of proportions showed a significant difference (χ^2^ = 12.048, df = 5, *p* = 0.034; Figure 5), the Holm–Bonferroni-adjusted *p*-values did not show significant differences between the proportion of introduced/native species over the 30–year periods (Figure 6)**.**

### 3.4. Invasiveness Risk Assessment for Alien Plant Species

Following the GCW 2017 risk classification [58] allowed the invasiveness risk for the 106 alien plants found in the archipelago to be assigned as follows: 18 species are considered to present an extremely high risk, 18 species represent a high risk, 32 species a medium risk, and 22 species present a low risk of invasiveness. Although eleven species found in the Revillagigedo are included in the GCW 2017, the information or number of reports necessary to determine their risk of invasiveness is not available; five more species are not included in the compendium (Table 3, Figure 7).

### 3.5. Comparison of the Status of Native and Alien Plant Species of Different Insular Regions

The GLM model derived from the quasi-Poisson family shows that smaller insular regions, on average, show a 1:1 ratio between native and introduced species, while in larger insular regions, native species exceed introduced ones, on average (Figure 8). According to this model, area is a good predictor of the number of introduced and native species on insular regions; however, the analyses of residual deviance suggest that other variables should be considered (χ^2^ = 400.9, df = 9, *p* ≤ 0.001, for introduced species; χ^2^ = 1482.2, df = 9, *p* ≤ 0.001, for native species) (Table 4).

## 4. Discussion

### 4.1. Inventory Overview

The impact that invasive plant species may have on the native flora and fauna of the Revillagigedo Archipelago requires a complete assessment. Invasive plants represent a potential threat to the survival of endangered bird species, such as the endemic Socorro Mockingbird (*Mimus graysoni*) and Socorro Parakeet (*Psittacara brevipes*), and other endemic and native species, because these plants can take over key areas required for their breeding success. Invasive plants may also jeopardize reintroduction efforts and the establishment of the Socorro Dove. There are numerous examples of the risks posed by invasive plants to island ecosystems. For instance, in the Galapagos Islands, where introduced plant species outnumber native species [25], many of the invasive species, particularly trees, vines, and grasses, compete aggressively with native plants, displacing them and altering a globally unique habitat, home to numerous endemic species [72]. The Hawaiian Archipelago, rich in endemic animals and plants, is also one of the ecosystems most threatened by invasive plants [73]. On these islands, once-abundant plant species are now threatened, and native habitats are dominated by highly invasive exotic plants that are capable of dramatically altering community structure and ecosystem processes [73]. In other archipelagos and islands, invasive plants are one of the main threats to endemic species and plant diversity, such as the Azores Islands, Madeira and the Canary Islands in the Atlantic Ocean, the Juan Fernandez Archipelago in the south of the Pacific Ocean, and the Mascarene Islands in the southwest of the Indian Ocean [74].

Knowledge on the flora of the Revillagigedo Archipelago has grown significantly over time. Johnston [19] provided the first account that included a list for the Revillagigedo Archipelago; he derived this list from collecting trips carried out between 1839 and 1925. More than five decades later, Levin and Moran [1] published a more complete inventory, which included specimens collected between l957 and 1988. They listed 117 native and 47 naturalized species for Socorro, while 41 native and one naturalized species were listed for Clarion. Several inventories have been published since, for instance, the management programs for the Revillagigedo Archipelago Biosphere Reserve [36] and the Revillagigedo National Park [37]. These inventories used scientific publications, internal reports, and databases such as EncicloVida [38].

More recently, Rebman et al. [20] reported 201 plant species for Socorro, of which 67 were considered non-native. On Clarion, they reported 58 species, of which 12 were considered non-native. On San Benedicto, they reported eleven species and no alien plants. Rebman et al. [20] considered 12 alien species as native, while other authors considered them introductions because of the date of first collection and closeness to human settlements (e.g., Levin and Moran [1]). Also, they claimed to report 38 “new” records (21 for Socorro, 16 for Clarion, and 1 for San Benedicto); however, 11 of these records were reported or collected during previous expeditions. This highlights the need for a thorough revision of Mexican and international herbaria when attempting to build a species inventory.

Since Levin and Moran’s flora, 57 additional species have been introduced to Socorro and 14 to Clarion. We report three new records for Socorro: *Cassia fistula*, which was assigned a *Cultivated* status; *Senna alata*, with a *Casual* status, and *Urochloa maxima*, which was found to be already *Naturalized*. It should be noted that these three species belong to families known for their high invasive potential (Fabaceae and Poaceae) [49], and it is imperative to establish immediate control actions to prevent their spread throughout the island. We also report, for the first time, four ornamental introduced species that have been kept under human care in the naval garrison: *Agave americana*, *Aloe vera*, *Cycas revoluta*, and *Ficus benjamina*. Although, for the moment, these plants do not pose a significant invasiveness risk, we recommend that they be monitored and considered examples of the continuous introduction of alien plant species into the island.

*Leucaena leucocephala* is the most common invasive species worldwide. This legume grows in the form of a tree or shrub, with high tolerance to drought and nitrogen-fixing root nodules, and thrives in arid and nutrient-poor environments [49]. The impact of *L. leucocephala* as a weed around the world is well known [58]. In the Hawaiian Archipelago, this species has spread quickly, invading lowland plant communities and hindering the proliferation of native species. By 2014, this species had already fully covered the lowlands of O’ahu and Kaua’i, as well as large lowland areas in the rest of that archipelago [64]. In Galapagos, the dissemination and impact of *L. leucocephala* has been less dramatic, but its presence has been reported in agricultural areas of Floreana, Isabela, San Cristobal, and Santa Cruz Islands, and plants have already been detected in the National Park area [75]. In the South Pacific, the Vanuatu islands have large extensions of monodominant shrubs of this invasive species [76]. In their global analysis of naturalized alien flora, Pyšek et al. [77] also found that *L. leucocephala* was present with an invasive status in 29.5% of the 349 regions (mainland and islands) considered in the study, of which 55 were Pacific Islands. On Socorro, this species was first registered by our team in 2015 and its continuous presence confirmed by Martínez-Gómez [31] and Rebman et al. [20]. *Leucaena leucocephala* has the highest GCW score in the Revillagigedo Archipelago.

Another invasive species of tree posing a high risk to Socorro native plants and endemic birds is the guava tree (*Psidium guajava*). This species, whose presence on Socorro was first reported by Cruz-Cisneros in 1967 (Appendix A), has numerous reports on its invasiveness around the world [58,77]. This species is considered one of the most damaging invasive plants in the Galapagos Archipelago [25], where it has become the most abundant invasive species in rural areas [62], and has displaced entire endemic tree forests such as *Scalesia pedunculata* [76].

Some herb species also stand out for their extreme invasiveness risk. *Senna obtusifolia* is a Fabaceae species with almost 200 reports of its invasiveness from around the world, including islands in the Caribbean, Pacific, and Indian Oceans [58]. This species threatens the ecosystems by competing with vulnerable native species and forming monospecific stands [78]. In Socorro, it can be observed covering large patches between 300 and 350 m in the southeastern region [31]. However, it is the grasses that have most successfully colonized the areas devoid of their original vegetation cover. *Melinis repens* is a fast-growing pioneer grass with the potential to colonize highly disturbed sites [79,80] and that has taken advantage of the bare and eroded soil areas on Socorro. On Desecheo Island, Puerto Rico, García-Cancel et al. [81] found that from 2000 to 2016, this grass went from being nearly absent to being present in almost all exposed areas of the island. In Socorro, Walter and Levin [9] described the rapid spread of *M. repens* in the south end of the island since its establishment in the late 1970s. Recently, Martínez-Gómez [31] confirmed that this species, along with *Cenchrus ciliaris* [20], had become the dominant cover in the most disturbed open areas. *C. ciliaris* poses an extreme invasiveness risk [58] and it is considered a threat to native plant communities in many islands of the world because it forms extensive, dense, and low-diversity stands, which inhibit the growth of native plant species; it also promotes frequent and intense fires [81,82,83], which have already occurred on occasion near the naval base on Socorro (Martínez-Gómez, unpublished field notes).

Worldwide, the families Asteraceae, Poaceae, and Fabaceae are the most represented in naturalized floras [77]. Insular ecosystems also show this pattern. Kueffer et al. [49] found, in 30 insular regions in the Atlantic, Caribbean, Pacific, and Western Indian Oceans, that Fabaceae, Poaceae, and Asteraceae were the best represented. In the Eastern Pacific, Guézou et al. [62] found that Fabaceae, Asteraceae, and Poaceae were also the most common in the Galapagos. On Socorro and Clarion, we observed that Fabaceae, Poaceae, and Malvaceae were the most common families.

### 4.2. Origin and Life History

The percentage of naturalized species native to North America (65%) found in Revillagigedo is higher than the 17% found among alien species with this origin in mainland and island regions around the world reported by Pyšek et al. [77]. This could be the result of the proximity of this continent to the archipelago. Furthermore, naval personnel and their families most likely introduced, intentionally or unintentionally, species that were familiar and closer to them [15]. Thus, only 11 of the 43 naturalized species native to North America were collected before the first naval garrison was established on Socorro in 1957 and only one was reported for Clarion before the establishment of the naval garrison in 1979. Species native to Temperate and Tropical Asia in the Revillagigedo Archipelago are present in higher percentages than those for the global naturalized flora, reported by Pyšek et al. [77], with 37.9% and 28.8% compared with 20.3% and 11.7%, respectively. We also found a greater representation of African species, with 35%, compared with the global proportion of 14.3%; this is most likely a result of the high number of African grasses [84] and other ruderal species, which, once introduced, took advantage of the overgrazed and eroded areas, rapidly colonizing the bare soil [81,85].

Regarding species’ life history (life form and habit), herbs were the best represented in the Revillagigedo Archipelago, where 53% of all naturalized species are herbs (30.3% annuals and 22.7% perennials). This finding is coincident with the trend observed in Galapagos by Guézou et al. [62], where herbs accounted for almost half of the alien species found in the archipelago. At a global level, the best-represented life histories among naturalized flora on island regions are perennial herbs, composing more than one third of all species (33%), while annual herbs represent 19%. Annual and perennial grasses in the Revillagigedo Archipelago are present in a higher percentage than the global trend at 15% and 9%, against 2.5% and 4.6%, respectively. It is noticeable that Revillagigedo’s percentage of annuals (47%) doubles the 22% found in other island regions, while the proportion of perennial woody species is lower at 16.7% compared to the 34.4% found on islands worldwide [77]. When compared with the percentages of oceanic islands only, Revillagigedo’s perennial woody species percentage is also lower than the 57% reported by Kueffer et al. [49], as well as the percentage of perennials (53%), which is lower than the 93% reported by these authors.

### 4.3. Year of Introduction

Th results from the comparisons between native/introduced species proportions over time showed that on Socorro, the proportion of introduced species became significantly higher in the years after the establishment of naval facilities. However, Clarion did not present a significant difference in these proportions after the establishment of the naval garrison, possibly as an effect of the small total number of species present on the island. Nevertheless, these results emphasize the need to reconsider and reinforce the biosecurity practices carried out in the archipelago.

It is necessary to highlight that, among the 106 introduced species included in our list (including species under cultivation), there are 19 species whose first reports are subsequent to the publication of the management program of the Biosphere Reserve in 2007, despite the explicit prohibition to introduce plants. This finding is particularly alarming since, among the species that have entered and become established after this prohibition, there are some that represent an extreme risk to the ecosystem due to their high potential for invasiveness, and, therefore, require immediate actions of control and eradication; these species include *Leucaena leucocephala*, *Ricinus communis* [86], and *Tridax procumbens* [20]. The extreme invasiveness of *R. communis* has also been reported for numerous countries around the world [58]. In fact, Pyšek et al. [77] found that this species is the second most widely distributed naturalized species on the planet, being recorded as an invader in 259 of 483 mainland regions and 161 of 361 island regions considered in their study, with 45 of the latter being located in the Pacific Ocean. On the Galapagos Archipelago, this species is considered one of the worst invaders [87], while *Tridax procumbens*, being one of the 200 most invasive plant species in the world, was found to be reported as invasive on 28 Pacific islands.

### 4.4. Invasiveness Risk Assessment for Alien Plant Species

Even when the minimum residency time for the invasion status classification was adapted so it would be relevant to the fragility of these islands and the characteristics of the alien plant species present on the archipelago, the mere fact of being able to classify the invasion status for some species such as *Leucaena leucocephala* or *Ricinus communis* may not be enough to carry out timely control actions. This is the case because, as soon as naturalization is achieved, the impacts may be both immediate and costly [58]. Taking this into account, we believe that by using weed risk assessment systems, like the Global Compendium of Weeds, that allow us to extrapolate knowledge about the invasion success of species in other areas of the world, we can advance to a prediction of the possible effects of the species in these new areas [26]. Nevertheless, we agree that this approach must not be taken at a distance without addressing the situation in the field, because the invasion history of each species on the Revillagigedo Archipelago itself should also be considered as a determinant for management actions. For example, *Melia azedarach* and *Psidium guajava* present the same score and risk category; however, the history of the presence of *P. guajava* on Socorro is older, its distribution much broader, and the risk of dispersal to less disturbed areas greater. Complementing the categorization of the invasiveness risk of the species with records of their invasion success on the island in question will thus provide a more complete framework for decision making. Therefore, we implemented a pilot control program based on mechanical methods for *Leucaena leucocephala*, *Psidium guajava*, *Acacia farnesiana*, *Citrus aurantium*, *Pithecellobium dulce*, *Cedrela odorata*, *Swietenia humilis* and *Ricinus communis*; preliminary results suggest that chemical methods should be incorporated to outcompete species with strong vegetative growth [86].

### 4.5. Comparison of the Status of Native and Alien Plant Species of Different Insular Regions

Less than 40% of the flora in the Revillagigedo Archipelago is alien, which is a percentage lower than those reported for other islands or archipelagos. For example, Pyšek et al. [77] found that, for 41% of the island regions included in their global study, naturalized alien plants accounted for more than 40% of their flora. Nevertheless, Socorro, which was the first island in Revillagigedo to be inhabited, is slowly approximating the trend presented by most inhabited oceanic islands, where there is practically a one-to-one ratio between the number of native and naturalized species [88]. There are several examples of this phenomenon in the Pacific Ocean. For the Galapagos Islands, 754 introduced species are registered (60% of an approximate total of 1254 species) [62]; of the total of these introduced species, about 45% are considered naturalized, and, of these species, 7% are considered invasive and an additional 20% potentially invasive [25]. In the Hawaiian archipelago, the number of naturalized species is around 1100 (48% of an approximate total of 2307 species) [64]. French Polynesia and the Cook Islands are also close to this 1:1 introduced/native ratio at 0.75 and 0.97, respectively, while the Juan Fernández Archipelago has already surpassed it with a ratio of 1.1. Examples from the Indian Ocean show a similar scene. La Reunion presents a 0.93 ratio, while the Seychelles and Mauritius surpass the 1:1 ratio at 1.06 and 1.07, respectively [24].

In our exploration of the relationship between the size of islands or island groups and the number of native and introduced plant species, we found that insular regions with smaller areas were the ones closer to this 1:1 ratio. However, there are other variables that may be involved in the native and introduced species richness. Other studies have identified size of the native flora, human population size [65], human development (measured by the gross domestic product (GDP) per capita) [49,57,77], habitat heterogeneity [61,89], geographic region and altitude [77] as significant contributors to the number of alien species. Among the reasons why the Revillagigedo Archipelago percentages of introduced species are lower than the ones presented in other tropical oceanic islands, despite human occupancy, may be the late colonization and the lack of a permanent human population relying on the islands’ resources.

## 5. Conclusions

This is the first work to curate and gather information from botanical material collected more than a century ago, located at Mexican and international institutions. This resulted in a complete and accurate list of the vascular plant species present in the Revillagigedo National Park, substantiated by an extensive review of herbaria. Certainly, additional curatorial work and taxonomic revisions will change the inventory composition. An awareness of the presence and risk posed by the steadily increasing number of alien species is crucial to propose and carry out timely control or eradication actions [89], thereby optimizing costs and resources as part of a restoration program on these Mexican islands. *Leucaena leucocephala*, *Psidium guajava*, *Ricinus communis*, and *Cenchrus ciliaris* are among the species that require immediate control actions. Constant monitoring, control efforts, and regular botanical surveys are also necessary to maintain the number of alien species in the Revillagigedo Archipelago. Control and eradication measures for alien plants will not only benefit engendered species such as the Socorro Mockingbird and Socorro Parakeet, but will also facilitate the reintroduction of the Socorro Dove.

## Figures and Tables

**Figure 1 plants-12-03455-f001:**
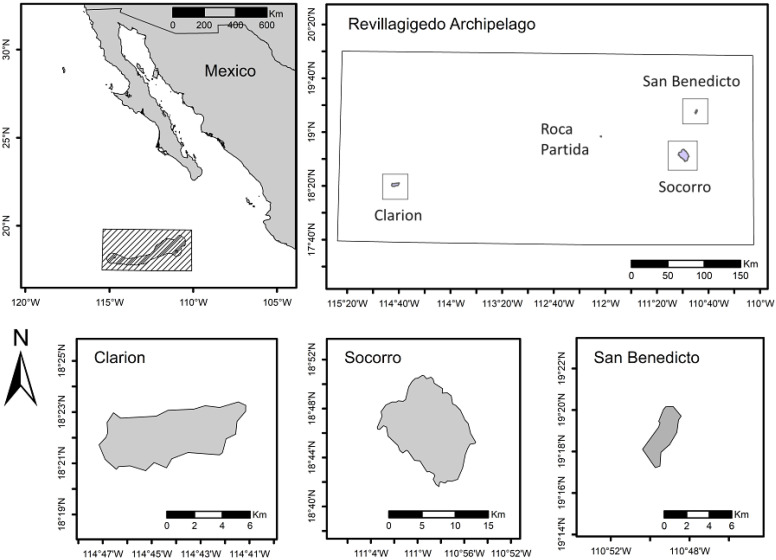
Location of the Revillagigedo Archipelago National Park in the Mexican Pacific Ocean, in relation to the mainland, and location of the three main islands.

**Figure 2 plants-12-03455-f002:**
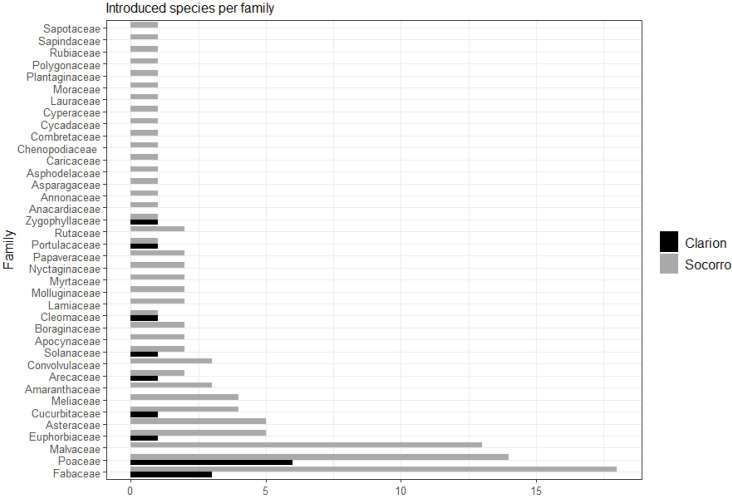
Number of alien plant species by family present in the Revillagigedo Archipelago.

**Figure 3 plants-12-03455-f003:**
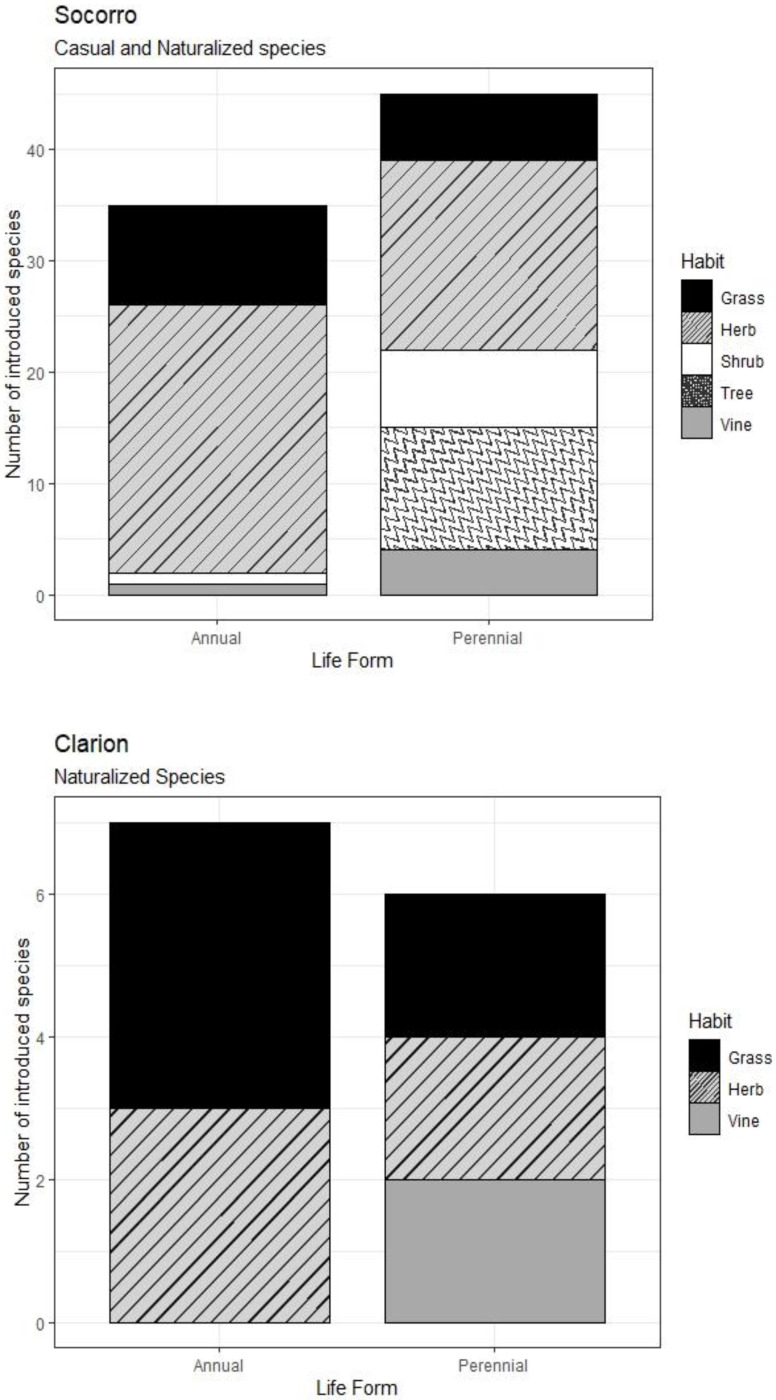
Habit and life form of casual and naturalized plant species on Socorro and naturalized plant species on Clarion.

**Figure 4 plants-12-03455-f004:**
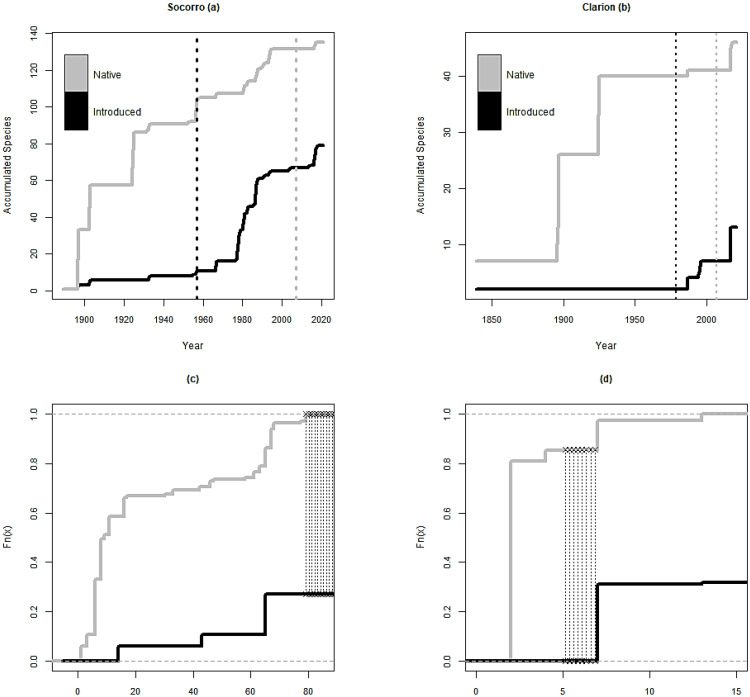
Species accumulation curves of native and introduced plant species for (**a**) Socorro and (**b**) Clarion. Black dotted lines mark the year in which the naval garrison was established on each island and gray dotted lines mark the year when the introduction of alien species was prohibited by the management program of the Biosphere Reserve [36]. Kolmogorov–Smirnov two-sample test plots: thin black dotted lines show the maximum difference between the two cumulative distributions for (**c**) Socorro and (**d**) Clarion.

**Figure 5 plants-12-03455-f005:**
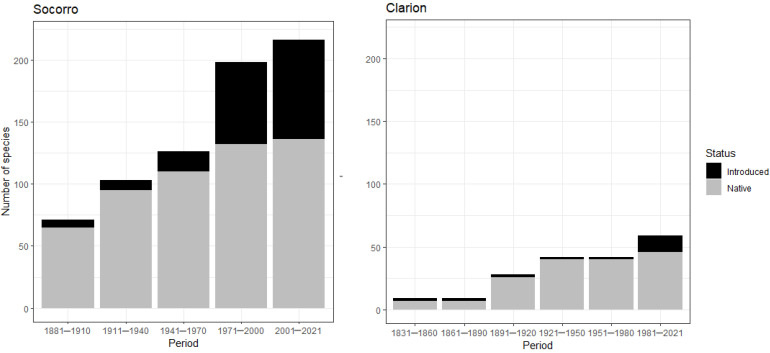
Total number of native and introduced (casual and naturalized) plant species recorded on Socorro and Clarion, for 30–year periods, from the year of the first species reported to 2021.

**Figure 6 plants-12-03455-f006:**
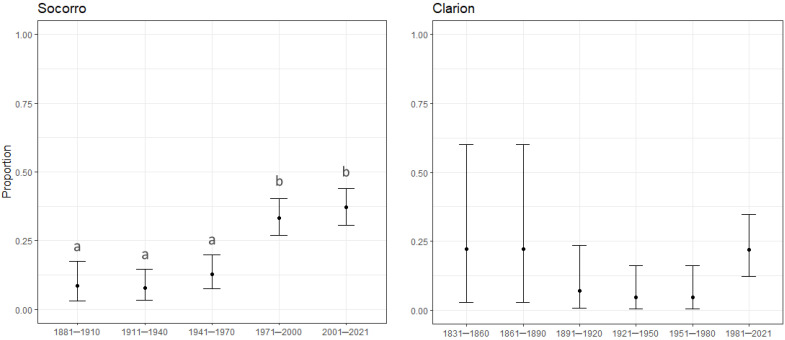
Proportion of introduced species by total number of species known at each 30–year period on Socorro and Clarion, from the year when the first species was reported on each island to 2021. Ninety-five percent confidence intervals were derived from a binomial distribution. In Socorro, the Holm–Bonferroni adjustment was able to separate the 30–year periods in two significantly different groups, marked with a and b in the figure. Although at least one significant pairwise difference between proportions on Clarion was detected by the multiple sample test for equality of proportions, the Holm–Bonferroni adjustment failed to identify it.

**Figure 7 plants-12-03455-f007:**
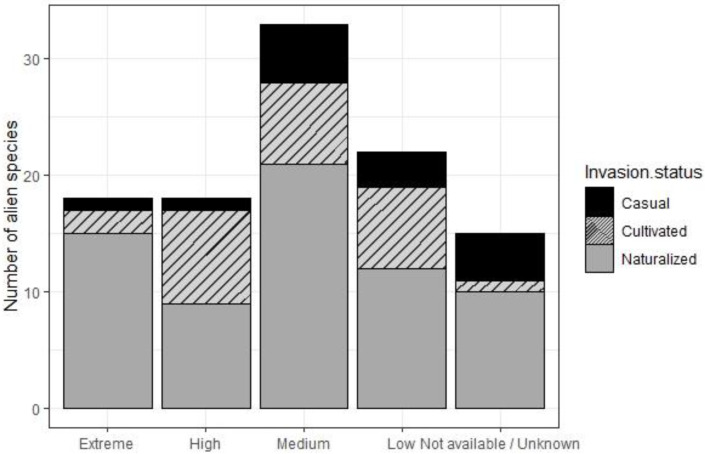
Total number of alien plant species by invasion status and GCW 2017 invasiveness risk rating categories in the Revillagigedo Archipelago.

**Figure 8 plants-12-03455-f008:**
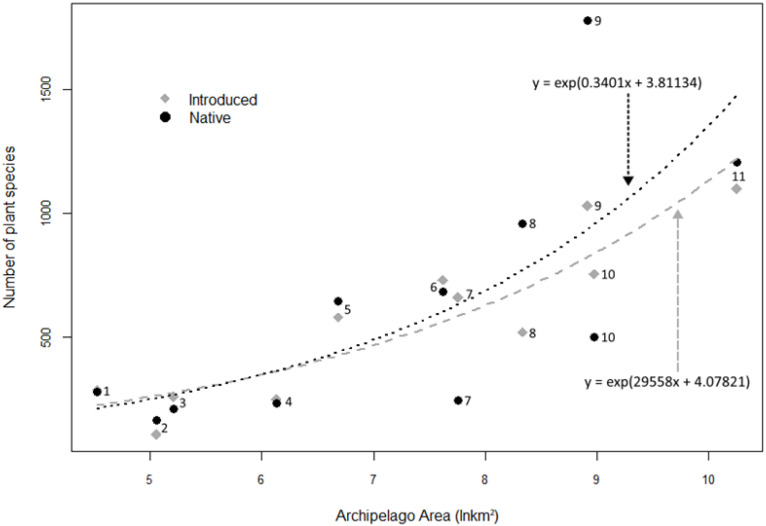
Number of introduced and native plant species from eleven insular regions around the world, as follows: (1) Cook, (2) Revillagigedo, (3) Juan Fernandez, (4) Seychelles, (5) Madeira, (6) Mauritius, (7) Azores, (8) French Polynesia, (9) Canaries, (10) Galapagos, (11) Hawaii. GLM based on the quasi-Poisson family are depicted: native species by black dots and invasive species by the gray broken line.

**Table 1 plants-12-03455-t001:** Revillagigedo Archipelago vascular plant species inventory. S = Socorro, C = Clarion, SB = San Benedicto. (*) Symbol marks whether the species is Endemic or Introduced and/or Probably Not Present; according to each column.

Family	Scientific Name	Island	Endemic	Introduced	Probably Not Present
Acanthaceae	*Elytraria imbricata* (Vahl.) Pers.	S			
Aizoaceae	*Sesuvium portulacastrum* (L.) L.	C			
Amaranthaceae	*Amaranthus palmeri* S. Watson	S		*	
Amaranthaceae	*Gomphrena nitida* Rothr.	S		*	
Amaranthaceae	*Gomphrena serrata* L.	S		*	
Amaranthaceae	*Iresine diffusa* Humb. & Bonpl. ex Willd.	C			
Anacardiaceae	*Mangifera indica* L.	S		*	
Annonaceae	*Annona muricata* L.	S		*	
Apiaceae	*Daucus montanus* Humb. & Bonpl. ex Schult.	S			
Apocynaceae	*Cascabela thevetia* (L.) Lippold	S		*	
Apocynaceae	*Metastelma californicum* Benth.	S			
Apocynaceae	*Metastelma cuneatum* Brandegee	S			
Apocynaceae	*Metastelma minutiflorum* Wiggins	S			
Aquifoliaceae	*Ilex socorroensis* Brandegee	S	*		
Araliaceae	*Oreopanax xalapensis* (Kunth) Decne. & Planch.	S			
Arecaceae	*Cocos nucifera* L.	S, C		*	
Arecaceae	*Phoenix dactylifera* L.	S		*	
Aristolochiaceae	*Aristolochia islandica* Pfeiffer	SB	*		*
Aristolochiaceae	*Aristolochia socorroensis* Pfeifer	S	*		
Aristolochiaceae	*Aristolochia* sp.	C	*		
Asparagaceae	*Agave americana* L.	S		*	
Asphodelaceae	*Aloe vera* (L.) Burm.f.	S		*	
Aspleniaceae	*Asplenium formosum* Willd.	S			
Aspleniaceae	*Asplenium sessilifolium* Desv.	S			
Asteraceae	*Ageratina pacifica* (B.L. Rob. ex B.L. Rob.) R.M. King & H. Rob.	S			
Asteraceae	*Ageratina pichinchensis* (Kunth) R.M. King & H. Rob.	S			
Asteraceae	*Ambrosia confertiflora* DC.	S		*	
Asteraceae	*Bahiopsis chenopodina* (E. Greene) E. E. Schill. & Panero	S			
Asteraceae	*Bidens socorrensis* Moran & G. A. Levin	S	*		
Asteraceae	*Brickellia peninsularis* Brandegee	S	*		
Asteraceae	*Brickellia peninsularis* Brandegee var. *amphithalassa* I. M. Johnst.	C	*		
Asteraceae	*Coreocarpus insularis* (Brandegee) E. B. Smith	S	*		
Asteraceae	*Eclipta prostrata* (L.) L.	S			
Asteraceae	*Erigeron crenatus* Eastwood ex I. M. Johnst.	SB	*		*
Asteraceae	*Erigeron socorrensis* Brandegee	S	*		
Asteraceae	*Gamochaeta sphacelata* (Kunth) Cabrera	S			
Asteraceae	*Gnaphalium attenuatum* DC.	S			
Asteraceae	*Laennecia confusa* (Cronq.) G. L. Nesom	S			
Asteraceae	*Perityle socorrosensis* Rose	S, C, SB	*		
Asteraceae	*Blumea viscosa* (Miller) D’Arcy	S		*	
Asteraceae	*Sonchus asper* (L.) Hill	S		*	
Asteraceae	*Sonchus tenerrimus* L.	S		*	
Asteraceae	*Tridax procumbens* L.	S		*	
Asteraceae	*Eremosis littoralis* Gleason	S	*		
Boraginaceae	*Cordia curassavica* (Jacq.) Roemer & Schultes	S			
Boraginaceae	*Cordia cylindrostachya* (Ruiz & Pav.) Roem. & Schult.	S		*	
Boraginaceae	*Heliotropium curassavicum* L.	S			
Boraginaceae	*Heliotropium curassavicum* L. var. *oculatum* (Heller) Thorne	C			
Boraginaceae	*Heliotropium procumbens* Miller	S		*	
Boraginaceae	*Tournefortia capitata* M. Martens & Galeotti	S			
Boraginaceae	*Tournefortia hartwegiana* Steud.	S			
Brassicaceae	*Lepidium lasiocarpum* Torr. & A. Gray var. *latifolium* C. L. Hitchc.	C			
Brassicaceae	*Lepidium virginicum* L. ssp. *pubescens* (Greene) C. L. Hitchc.	S			
Burseraceae	*Bursera epinnata* (Rose) Engl.	S			
Cactaceae	*Opuntia* sp. 1	C			
Cactaceae	*Opuntia* sp. 2	S			
Campanulaceae	*Calcaratolobelia cordifolia* (Hook. & Arn.) Wilbur	S			
Campanulaceae	*Triodanis perfoliata* subsp. *biflora* (Ruiz & Pav.) Lammers	S			
Caricaceae	*Carica papaya* L.	S		*	*
Caryophyllaceae	*Arenaria lanuginosa* (Michx.) Rohrb.	S			
Chenopodiaceae	*Dysphania ambrosioides* (L.) Mosyakin & Clemants	S		*	
Cleomaceae	*Arivela viscosa* (L.) Raf.	S, C		*	
Combretaceae	*Conocarpus erectus* L.	S			
Combretaceae	*Terminalia catappa* L.	S		*	
Commelinaceae	*Commelina erecta* L.	C			
Convolvulaceae	*Cressa truxillensis* Kunth	S, C			
Convolvulaceae	*Ipomoea carnea* subsp. *fistulosa* (Mart. ex Choisy) D.F. Austin	S		*	
Convolvulaceae	*Ipomoea halierca* I. M. Johnst.	C	*		
Convolvulaceae	*Ipomoea imperati* (Vahl) Griseb.	S			
Convolvulaceae	*Ipomoea indica* (Burm. f.) Merrill	C			
Convolvulaceae	*Ipomoea pes-caprae* (L.) R. Br.	S, C, SB			
Convolvulaceae	*Ipomoea purpurea* (L.) Roth	S		*	
Convolvulaceae	*Ipomoea triloba* L.	S			
Convolvulaceae	*Distimake quinquefolius* (L.) A.R Simões & Staples	S		*	
Cucurbitaceae	*Citrullus lanatus* (Thunb.) Matsum. & Nakai.	S		*	*
Cucurbitaceae	*Lagenaria siceraria* (Mol.) Standl.	S		*	*
Cucurbitaceae	*Luffa cylindrica* (L.) M. Roem.	S, C		*	*
Cycadaceae	*Cycas revoluta* Thunb.	S		*	
Cyperaceae	*Bolboschoenus maritimus* (L.) Palla ssp. *paludosus* (A. Nelson) T. Koyama	C			
Cyperaceae	*Bulbostylis nesiotica* (I. M. Johnst.) Fern.	S, C, SB	*		SB *
Cyperaceae	*Cyperus duripes* I. M. Johnst.	S, C, SB	*		
Cyperaceae	*Cyperus ligularis* Hemsl.	S			
Cyperaceae	*Cyperus rotundus* L.	S		*	
Cyperaceae	*Cyperus sordidus* J. Presl & C. Presl	S			
Cyperaceae	*Eleocharis mutata* (L.) Roem. & Schult.	C			
Dennstaedtiaceae	*Pteridium caudatum* (L.) Maxon	S			
Dryopteridaceae	*Ctenitis equestris* (Kunze) Ching	S			
Dryopteridaceae	*Dryopteris knoblochii* A. R. Smith	S			
Dryopteridaceae	*Polystichum hartwegii* (Klotzsch) Hieron	S			
Euphorbiaceae	*Acalypha umbrosa* Brandegee	S	*		
Euphorbiaceae	*Croton masonii* I. M. Johnst.	S	*		
Euphorbiaceae	*Euphorbia anthonyi* (Brandegee) G. A. Levin var. *anthonyi*	S	*		
Euphorbiaceae	*Euphorbia anthonyi* (T.S. Brandegee) G. Levin	SB	*		
Euphorbiaceae	*Euphorbia anthonyi* Brandegee var. *clarionensis* (Brandegee) I. M. Johnst.	C	*		
Euphorbiaceae	*Euphorbia californica* Benth.	C			
Euphorbiaceae	*Euphorbia heterophylla* L.	S		*	
Euphorbiaceae	*Euphorbia hirta* L. var. *hirta*	S		*	
Euphorbiaceae	*Euphorbia hyssopifolia* L.	S		*	
Euphorbiaceae	*Euphorbia incerta* Brandegee	S			
Euphorbiaceae	*Euphorbia thymifolia* L.	S, C		*	
Euphorbiaceae	*Hippomane mancinella* L.	S			
Euphorbiaceae	*Ricinus communis* L.	S		*	
Fabaceae	*Guilandina bonduc* L.	S, C			
Fabaceae	*Canavalia rosea* (Sw.) DC.	S, C			
Fabaceae	*Cassia fistula* L.	S		*	
Fabaceae	*Crotalaria incana* L.	S		*	
Fabaceae	*Delonix regia* (Hook.) Raf.	S		*	
Fabaceae	*Desmanthus bicornutus* S. Watson	S		*	
Fabaceae	*Desmodium procumbens* (Mill.) Hitchc.	S		*	
Fabaceae	*Desmodium scorpiurus* (Sw.) Desf.	S		*	
Fabaceae	*Galactia striata* (Jacq.) Urban	C			
Fabaceae	*Indigofera suffruticosa* Miller	S		*	
Fabaceae	*Leucaena leucocephala* (Lam.) de Wit	S		*	
Fabaceae	*Macroptilium atropurpureum* (DC.) Urban	S, C		*	
Fabaceae	*Neptunia plena* (L.) Benth.	S, C		*	
Fabaceae	*Phaseolus lunatus* L.	S			
Fabaceae	*Pithecellobium dulce* (Roxb.) Benth.	S		*	
Fabaceae	*Prosopis juliflora* (Sw.) DC.	S		*	
Fabaceae	*Rhynchosia minima* (L.) DC.	S, C		*	
Fabaceae	*Senna alata* (L.) Roxb.	S		*	
Fabaceae	*Senna obtusifolia* (L.) Irwin & Barneby	S		*	
Fabaceae	*Sophora tomentosa* L.	C			
Fabaceae	*Tamarindus indica* L.	S		*	
Fabaceae	*Vachellia campechiana* (Mill.) Seigler & Ebinger	S		*	
Fabaceae	*Vachellia farnesiana* (L.) Wight & Arn. var. *farnesiana*	S		*	
Fabaceae	*Zapoteca formosa* subsp. *rosei* (Wiggins) H.M. Hern.	S, C	*		
Gentianaceae	*Centaurium capense* C. R. Broome	S			
Gentianaceae	*Zeltnera wigginsii* (Broome) Mansion	S			
Goodeniaceae	*Scaevola plumieri* (L.) Vahl	S			
Hypericaceae	*Hypericum eastwoodianum* I. M. Johnst.	S	*		
Lamiaceae	*Lepechinia hastata* (A. Gray) Epling ssp. *socorrensis* Moran	S	*		
Lamiaceae	*Hyptis pectinata* (L.) Poit.	S		*	
Lamiaceae	*Salvia pseudomisella* Moran & G. A. Levin	S	*		
Lamiaceae	*Salvia misella* Kunth	S		*	
Lamiaceae	*Teucrium townsendii* Vasey & Rose var. *affine* (Brandegee) Moran	S	*		
Lamiaceae	*Teucrium townsendii* Vasey & Rose var. *townsendii*	C	*		
Lamiaceae	*Teucrium affine* Vasey & Rose var. *dentosum* I.M. Johnst.	SB	*		*
Lauraceae	*Persea americana* Mill.	S		*	
Lycopodiaceae	*Huperzia dichotoma* (Jacq.) Trevis	S			
Malvaceae	*Abutilon californicum* Benth.	S			
Malvaceae	*Corchorus aestuans* L.	S		*	
Malvaceae	*Corchorus orinocensis* Kunth	S		*	
Malvaceae	*Gossypium hirsutum* L.	S		*	
Malvaceae	*Hibiscus diversifolius* Jacq.	S		*	
Malvaceae	*Hibiscus furcellatus* Desr.	S			
Malvaceae	*Talipariti tiliaceum* var. *pernambucense* (Arruda) Fryxell	S		*	
Malvaceae	*Malvastrum americanum* (L.) Torrey	S		*	
Malvaceae	*Malvastrum coromandelianum* (L.) Garcke	S		*	
Malvaceae	*Malvella leprosa* (Ortega) Krapov.	C			
Malvaceae	*Melochia pyramidata* L.	S, C		S *	
Malvaceae	*Pavonia hastata* Cav.	S			
Malvaceae	*Sida acuta* Burm.f.	S		*	
Malvaceae	*Sida barclayi* Baker f.	S		*	
Malvaceae	*Sida ciliaris* L.	S		*	
Malvaceae	*Sida nesogena* I. M. Johnst.	S	*		
Malvaceae	*Sida rhombifolia* L.	S		*	
Malvaceae	*Sida salviifolia* C. Presl	S		*	
Malvaceae	*Triumfetta socorrensis* Brandegee	S	*		
Malvaceae	*Waltheria indica* L.	S, C			
Meliaceae	*Cedrela odorata* L.	S		*	
Meliaceae	*Melia azedarach* L.	S		*	
Meliaceae	*Swietenia humilis* Zucc.	S		*	
Meliaceae	*Swietenia macrophylla* King	S		*	
Molluginaceae	*Glinus radiatus* (Ruíz López & Pavón) Rohrb.	S		*	
Molluginaceae	*Mollugo verticillata* L.	S		*	
Moraceae	*Ficus benjamina* L.	S		*	
Moraceae	*Ficus cotinifolia* Kunth	S			
Myrtaceae	*Psidium guajava* L.	S		*	
Myrtaceae	*Psidium socorrense* I. M. Johnst.	S	*		
Nyctaginaceae	*Boerhavia coccinea* Miller	S, C			
Nyctaginaceae	*Boerhavia erecta* L.	S		*	
Nyctaginaceae	*Boerhavia scandens* L.	C			
Nyctaginaceae	*Bougainvillea glabra* Choisy	S		*	
Oleaceae	*Forestiera rhamnifolia* Griseb.	S			
Onagraceae	*Oenothera resicum* Benavides, Kuethe, Ortiz-Alcaráz & León de la Luz	C	*		
Ophioglossaceae	*Botrychium socorrense* W.H. Wagner	S	*		
Ophioglossaceae	*Ophioglossum reticulatum* L.	S			
Orchidaceae	*Acianthera unguicallosa* (Ames & C. Schweinf.) Solano	S	*		
Orchidaceae	*Dendrophylax porrectus* (Rchb. f.) Carlsward & Whitten	S			
Orchidaceae	*Epidendrum nitens* Rchb. f.	S			
Orchidaceae	*Epidendrum rigidum* Jacq.	S			
Orchidaceae	*Guarianthe aurantiaca* (Bateman ex Lindl.) Dressler & W.E. Higgins	S			
Orchidaceae	*Sarcoglottis schaffneri* (Rchb. f.) Ames	S			
Orobanchaceae	*Castilleja bryantii* Brandegee var. *socorrensis* (Moran) J. M. Egger	S	*		
Papaveraceae	*Argemone ochroleuca* Sweet	S		*	
Passifloraceae	*Passiflora suberosa* L.	S			
Piperaceae	*Peperomia socorronis* Trel.	S	*		
Piperaceae	*Peperomia tetraphylla* (G. Forster) Hook. & Arn.	S			
Plantaginaceae	*Nuttallanthus texanus* (Scheele) D. A. Sutton	S			
Plantaginaceae	*Scoparia dulcis* L.	S		*	
Poaceae	*Aristida adscensionis* L.	S			
Poaceae	*Aristida tenuifolia* Hitchc.	C	*		
Poaceae	*Aristida vaginata* A. Hitchc.	S	*		
Poaceae	*Bothriochloa pertusa* (L.) A. Camus	S		*	
Poaceae	*Cenchrus ciliaris* L.	S, C		*	
Poaceae	*Cenchrus echinatus* L.	S, C		*	
Poaceae	*Cenchrus myosuroides* Kunth	S, SB			SB *
Poaceae	*Chloris barbata* Swartz	S, C		*	
Poaceae	*Cynodon dactylon* (L.) Pers.	S		*	
Poaceae	*Dactyloctenium aegyptium* (L.) Willd.	S, C		*	
Poaceae	*Digitaria bicornis* (Lam.) Roem. & Schult.	S		*	
Poaceae	*Echinochloa colona* (L.) Link	S		*	
Poaceae	*Eragrostis tenella* (L.) P. Beauv. ex Roem. & Schult.	S		*	
Poaceae	*Eragrostis ciliaris* (L.) Link.	S		*	
Poaceae	*Eragrostis prolifera* Vasey	SB			
Poaceae	*Eragrostis pectinacea* (Michx.) Nees	S		*	
Poaceae	*Eriochloa acuminata* (Presl). Kunth.	C			
Poaceae	*Heteropogon contortus* (L.) P. Beauv. ex Roem. & Schult.	S, C			
Poaceae	*Jouvea pilosa* (Presl) Scribn.	S			
Poaceae	*Leptochloa panicea* (Retz.) Ohwi	S		*	
Poaceae	*Melinis repens* (Willdenow) Zizka	S, C		*	
Poaceae	*Muhlenbergia solisii* (G. A. Levin) P. M. Peterson	S	*		
Poaceae	*Oplismenus hirtellus* (L.) Beauv.	S			
Poaceae	*Paspalum longum* Chase	S	*		
Poaceae	*Schizachyrium mexicanum* (Hitchc.) A. Camus	S			
Poaceae	*Schizachyrium sanguineum* (Retz.) Alston	S			
Poaceae	*Setaria geminata* (Forssk) Veldkamp	S			
Poaceae	*Setaria parviflora* (Poir.) Kerguelen	S			
Poaceae	*Sorghastrum pohlianum* Dávila, L.I. Cabrera & R. Lira	S	*		
Poaceae	*Sporobolus purpurascens* (Swartz) Hamilton	S			
Poaceae	*Sporobolus pyramidatus* (Lam.) Hitchc.	C			
Poaceae	*Urochloa maxima* (Jacq.) R.D. Webster	S		*	
Poaceae	*Urochloa reptans* (L.) Stapf	C		*	
Polygonaceae	*Antigonon leptopus* Hook. & Arn.	S		*	
Polypodiaceae	*Pecluma alfredii* (Rosenst.) M. G. Price var. *occidentalis* A. R. Smith	S			
Polypodiaceae	*Polypodium californicum* Kaulfuss	S			
Polypodiaceae	*Polypodium polypodioides* (L.) Watt. var. *aciculare* Weath.	S			
Portulacaceae	*Portulaca oleracea* L.	S, C		*	
Portulacaceae	*Portulaca pilosa* L	S, C			
Potamogetonaceae	*Potamogeton nodosus* Poiret	S			
Psilotaceae	*Psilotum nudum* (L.) P. Beauv.	S			
Pteridaceae	*Hemionitis radiata* (L.) Christenh.	S			
Pteridaceae	*Cheilanthes peninsularis* Maxon var. *insularis* Weath.	S, C	*		
Pteridaceae	*Pityrogramma ebenea* (L.) Proctor	S			
Rhamnaceae	*Frangula discolor* (Donn.Sm.) Grubov	S			
Rhamnaceae	*Rhamnus humboldtiana* Willd. ex Schult.	C			
Rosaceae	*Prunus serotina* Ehrh.	S			
Rosaceae	*Rubus* sp.	S			
Rubiaceae	*Chiococca alba* (L.) C. L. Hitchc.	S			
Rubiaceae	*Galium mexicanum* Kunth	S			
Rubiaceae	*Guettarda insularis* Brandegee	S	*		
Rubiaceae	*Houstonia mucronata* (Benth.) B.L. Rob.	S			
Rubiaceae	*Mitracarpus hirtus* (L.) DC.	S		*	
Rubiaceae	*Spermacoce nesiotica* (Robinson) G. A. Levin	S, C	*		
Rutaceae	*Citrus × limon* (L.) Osbeck	S		*	
Rutaceae	*Citrus x*× *aurantium* L.	S		*	
Rutaceae	*Zanthoxylum fagara* (L.) Sarg.	C			
Rutaceae	*Zanthoxylum insulare* Rose	S			
Sabiaceae	*Meliosma nesites* I. M. Johnst.	S	*		
Santalaceae	*Phoradendron quadrangulare* (Kunth) Griseb.	S			
Santalaceae	*Phoradendron tonduzii* Trel.	S			
Sapindaceae	*Cardiospermum halicacabum* L.	S, C			
Sapindaceae	*Dodonaea viscosa* (L.) Jacq.	S, C			
Sapindaceae	*Sapindus saponaria* L.	C			
Sapindaceae	*Sapindus* sp.	S		*	
Sapotaceae	*Manilkara zapota* (L.) P. Royen	S		*	
Sapotaceae	*Sideroxylon socorrense* (Brandegee) T. D. Penn.	S	*		
Solanaceae	*Cestrum pacificum* Brandegee	S	*		
Solanaceae	*Datura aff. discolor* Bernh.	C			
Solanaceae	*Nicotiana stocktonii* T. S. Brandegee	S, C, SB	*		
Solanaceae	*Physalis clarionensis* Waterf.	C	*		
Solanaceae	*Physalis mimulus* Waterf.	S	*		
Solanaceae	*Physalis philadelphica* Lam.	C		*	
Solanaceae	*Physalis pruinosa* L.	S		*	
Solanaceae	*Solanum americanum* Miller	S		*	
Solanaceae	*Solanum ferrugineum* Jacq.	S			
Thelypteridaceae	*Thelypteris oligocarpa* (Willd.) Ching	S			
Verbenaceae	*Citharexylum danirae* León de la Luz & F. Chiang	S	*		
Verbenaceae	*Lantana involucrata* L. var. *socorrensis* Moldenke	S	*		
Verbenaceae	*Lantana velutina* M. Martens & Galeotti	S			
Verbenaceae	*Verbena sphaerocarpa* L. M. Perry	S	*		
Zygophyllaceae	*Tribulus cistoides* L.	S, C		*	

**Table 2 plants-12-03455-t002:** Contribution of the world regions where alien species introduced into the Revillagigedo Archipelago are native, broken down by invasion status. Note that the total number exceeds the 104 species recorded in the archipelago because some species originate from more than one continent. Delimitation of regions followed the Biodiversity Information Standards Organization [47]. (-) Symbol indicates the absence of species in that category.

Region	Cultivated	Casual	Naturalized	Total
Africa	5	1	26	32
Asia Tem.	7	-	25	32
Asia Trop.	7	-	19	26
Australasia	4	-	9	13
Europe	-	-	5	5
N. Am.	11	12	43	66
S. Am.	9	11	38	58

**Table 3 plants-12-03455-t003:** Alien plant species in the Revillagigedo Archipelago with Extreme and High invasiveness risk according to the rating and score assigned by the Global Compendium of Weeds [58]. S = Socorro, C = Clarion, N = naturalized, Cas = casual, Cul = cultivated.

Family	Species	Island	Invasion Status	GCW Score	GCW Rating
Fabaceae	*Leucaena leucocephala* (Lam.) de Wit	S	N	64	Extreme
Fabaceae	*Pithecellobium dulce* (Roxb.) Benth.	S	Cas	51.2	Extreme
Fabaceae	*Vachellia farnesiana* (L.) Wight & Arn. var. *farnesiana*	S	N	51.2	Extreme
Cyperaceae	*Cyperus rotundus* L.	S	N	44.8	Extreme
Fabaceae	*Macroptilium atropurpureum* (DC.) Urban	S, C	N	44.8	Extreme
Fabaceae	*Senna obtusifolia* (L.) Irwin & Barneby	S	N	44.8	Extreme
Poaceae	*Cenchrus echinatus* L.	S, C	N	44.8	Extreme
Poaceae	*Cynodon dactylon* (L.) Pers.	S	N	44.8	Extreme
Portulacaceae	*Portulaca oleracea* L.	S, C	N	44.8	Extreme
Euphorbiaceae	*Ricinus communis* L.	S	N	43.2	Extreme
Arecaceae	*Phoenix dactylifera* L.	S	Cul	38.4	Extreme
Meliaceae	*Swietenia macrophylla* King	S	Cul	38.4	Extreme
Asteraceae	*Sonchus asper* (L.) Hill ssp. *asper*	S	N	35.84	Extreme
Fabaceae	*Crotalaria incana* L.	S	N	35.84	Extreme
Myrtaceae	*Psidium guajava* L.	S	N	34.56	Extreme
Poaceae	*Cenchrus ciliaris* L.	S, C	N	33.6	Extreme
Poaceae	*Dactyloctenium aegyptium* (L.) Willd.	S, C	N	33.6	Extreme
Poaceae	*Echinochloa colona* (L.) Link	S	N	33.6	Extreme
Arecaceae	*Cocos nucifera* L.	S, C	Cul	28.8	High
Asteraceae	*Tridax procumbens* L.	S	N	26.88	High
Malvaceae	*Sida acuta* Burm.f.	S	N	26.88	High
Chenopodiaceae	*Dysphania ambrosioides* (L.) Mosyakin & Clemants	S	N	26.4	High
Anacardiaceae	*Mangifera indica* L.	S	Cul	25.92	High
Fabaceae	*Cassia fistula* L.	S	Cul	25.92	High
Meliaceae	*Cedrela odorata* L.	S	N	25.92	High
Meliaceae	*Melia azedarach* L.	S	Cul	25.92	High
Convolvulaceae	*Ipomoea purpurea* (L.) Roth	S	Cul	24	High
Polygonaceae	*Antigonon leptopus* Hook. & Arn.	S	N	24	High
Solanaceae	*Solanum americanum* Miller	S	N	24	High
Poaceae	*Melinis repens* (Willdenow) Zizka	S, C	N	20.16	High
Caricaceae	*Carica papaya* L.	S	Cul	19.44	High
Combretaceae	*Terminalia catappa* L.	S	Cul	19.44	High
Euphorbiaceae	*Euphorbia heterophylla* L.	S	Cas	19.2	High
Zygophyllaceae	*Tribulus cistoides* L.	S, C	N	19.2	High
Poaceae	*Eragrostis amabilis* (L.) Wight & Arn.	S	N	17.92	High
Fabaceae	*Tamarindus indica* L.	S	Cul	17.28	High

**Table 4 plants-12-03455-t004:** Generalized linear model (GLM) for the number of introduced and native plant species versus archipelago or island surface using a quasi-Poisson model with a log link function.

	Estimate	SE	*t*-Value	*p*
Introduced				
Intercept	4.07821	0.41238	9.889	3.92 × 10^−6^
ln area (km^2^)	0.29558	0.04989	5.925	0.000222
Dispersion parameter: 42.16917				
Null deviance: 1989.76 (10 df)				
Residual deviance: 400.98 (9 df)				
Native				
Intercept	3.81134	0.81587	4.672	0.00117
ln area (km^2^)	0.3401	0.09741	3.491	0.00682
Dispersion parameter: 169.7447				
Null deviance: 3750.7 (10 df)				
Residual deviance: 1484.2 (9 df)				

## Data Availability

Data are available upon request.

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
