# Peer review of "Vascular Plant Species Inventory of Mexico’s Revillagigedo National Park: Awareness of Alien Invaders as a *Sine Qua Non* Prerequisite for Island Conservation"

_plants, 2023, doi:10.3390/plants12193455_

Round 1

Reviewer 1 Report

The presented manuscript is a quite well written study. It can be published after minor corrections. My comments are below.

At first, the title looks too large, extensive. I recommend to shorten a little it.

Concerning Introduction, I recommend to pay more attention to highlighting of the relevance of research related to invasive alien plants in and outside Protected Areas. It would increase the level of the paper.

In the last paragraph (lines 96-102), the authors should state the aim and research tasks of this study. But now it consists data, describing the results (what is presented here). 

The section Material and Methods is well written. All materials and research design are well described. My compliments.

In the section Results, the authors presented the obtained results with a large number of illustrations. However, I suggest to add more statistical treatments. Maybe, the correlation analysis can be applied here. But, in general, this section is good. Just minor corrections are needed. For instance, Table 1 is recommended to place in the Supplement Material.

In the section Discussion, the authors well explained and discussed the data presented in the section Results. The authors well used an amount of the literature to discuss the results. I think that this section is well written.

Finally, I recommend to add the section Conclusions, where the authors should state the main research implications and concluding remarks on the basis of the results presented and discussed.

Author Response

Reviewer 1

We consider that the length of this article corresponds to the depth with which we address the subject, and we only included relevant and necessary information.

Introduction:  We attended the recommendation by highlighting the relevance of research related to invasive alien plants in the Revillagigedo Archipelago, as well as other insular areas of the world. Now, the aim and research tasks of this study are stated in the last paragraph.

Results: We consider that the inclusion of Table 1 in the Results section is a priority for this article, since it is the main result. Moreover, this inventory will become a useful tool for conservationists working in the Revillagigedo Archipelago and also will increase the rate for citations for this article. In fact, we expect that our article will become a relevant citation for the following 20 years.

The suggestion of further statistical analyses was attended by including a new analysis to compare statistical trends between different insular regions of the world. We evaluated the relationship between the number of species versus island area for both native and alien plants using Generalized Linear Models.

Conclusions: We added a general conclusion summarizing the relevance of this work and the main implications.

Reviewer 2 Report

Please refer to the accompanied reviewer's annotated manuscript for editorial, language and scientific corrections to be made.

The quality of English can be improved. An online grammar checker found 241 issues in need of correction.

Some of the most notable issues include:

1. Irrelevant information in Introduction section

2. Some information in Methodology need source citations, also some information irrelevant.

3. Some results were not discussed.

4. Over citation of information - one fact = one source listing

6. Add the same sub-headings (see below) as used in the results section to: ·         Align the two sections.

·         Preserve the sequence of reporting

·         Ensure easy orientation for readers of paper

·         Ensure weightings are correctly balanced

·         Continuously refer back to Tables and Figures for reference purposes 

7. Origin and life history - Discussed?

8. Year of introduction - Discussed?

Author Response

Reviewer 2

The quality of English was improved. We had our manuscript checked by native English-speaking colleagues, who made all the pertinent grammar and spelling corrections.

Title: Tilte was changed to -Vascular plant species inventory of the Revillagigedo National Park: Awareness of alien invaders as a sine qua non prerequisite for island conservation-, in order to include our work with native plant species, and not only introduced species.

Abstract: We stated that the constant introduction of potentially invasive alien plant species and the lack of adequate control or eradication actions, jeopardize the conservation and restoration of Revillagigedo’s fragile ecosystems.

Keywords: Keywords were changed, so that elements included in the title were not repeated in this section and appeared alphabetically.

Introduction: We consider that there is no irrelevant information in the Introduction section.

All the references included in the citations of this section are relevant; they not only show an expert command of the subject but show historical trends as well. Since we dealt with historical aspects related to the compilation of the flora, we decided to leave all pertinent references, also, previously missing citations were added.

Description of Weed Risk Assessment systems was removed from this section and moved to the Materials and Methods section.

Methodology: Figure 1 was made clearer by including each island’s name and squares indicating which ones were magnified.

We specified the dominant grass species that typifies the “grassland” vegetation association.

 Previously missing citations were added.

We consider that the information about the endemic fauna of each island is relevant, since the Revillagigedo Archipelago is recognized by the Alliance for Zero Extinction for the presence of several endangered and critically endangered species. The survival of endemic and endangered fauna on each island is strongly dependent on habitat restoration and control of invasive plant species.

Results: The number of species “Not present but previously reported in Socorro Island” was corrected, and it was stablished that the remaining four species have not been observed since 2010.

We consider that with the changes to the title, it was made clear that our study encompasses all vascular plant species in the Archipelago, so, the inclusion of results regarding native and endemic species is necessary and pertinent.

Year of Introduction: Figure 4 was rearranged, so a), b), c) and d) are mentioned in order.

Discussion: We believe that all citations in this section are relevant, even more when comparing our results with trends in other insular ecosystems.

Information was reordered using the same subtitles as in the Results section.

Origin and life history and Year of introduction subsections are discussed, and comparisons were made whenever data from other insular areas were available.

Reviewer 3 Report

This study presents a complete list of plant species and an updated list of alien plant species as a primary tool for the restoration and conservation of the Revillagigedo Archipelago National Park, which was compiled through an extensive review of national and international plant collections and other sources. However, the impact that invasive plant species may have on the native flora and fauna of the Revillagigedo Archipelago requires a full assessment. In this regard, a list of 270 species was created that includes 106 alien plant species (104 in Socorro and 16 in Clarion), 67 (25%) are naturalized, 14 (5%) are occasionally alien, and 25 (8.5%) survive under cultivation. Alien species belong to 73 families and 64% of naturalized species are native to North America. Annual and perennial herbs are the predominant life forms in the alien flora of the archipelago (53%). As noted, the number of introduced species over time has increased significantly since the islands were inhabited, and many of the newly introduced species are considered highly invasive in other islands of the world. Therefore, this study adequately complements the species invasion risk categorization by providing a more complete framework for decision-making.

Author Response

Reviewer 3

Reviewer 3 did not require editions

Round 2

Reviewer 2 Report

For multiple citations in a sentence - place one of the citation numbers inside the sentence next to its associated fact, so that readers know which piece of information is associated with which author – correct this throughout the manuscriptstring citation is not allowed.

Figure 8 must be reported on and indicated in text before placement.

Table 4 must be reported on and indicated in text before placement.

On these islands, once abundant plant species are  now threatened; native habitats are currently (cannot state are now or currently and then cite papers from 2014 and 1997) dominated by highly invasive exotic plants, which are capable of dramatically altering community structure and ecosystem processes [74,75]

Nowadays (and then you cite 2014?) this species completely covers the lowlands of O’ahu and Kaua’i, as well as large lowland areas in the rest of that archipelago [67].

The Revillagigedo Archipelago still shows a larger number of native species in spite  of human occupancy (WHY?). Less than 40 % of the flora in the Revillagigedo Archipelago is alien; it is a lower percentage than those reported for other islands or archipelagos (WHY?). Many such instances of open statements without explanations occur in the discussion section - refer to the reviewer's annotated manuscript.

Surely your discussion cannot just consist of comparing to other studies (the so-called ”I-found-they-found” principle) – this type of reporting provides very limited context. You must present explanations as to WHY you found these results, and what are their impact? What conclusion can be drawn? If your results are in line with other studies – then what does the trend indicate, if different from other investigations, then why do yours differ from others? What recommendation can you make, what would be the impact of your recommendation? Etc. etc.

The quality of English is greatly improved from the previous version. Only minor issues persist, as indicated in the reviewer's annotated manuscript.

Author Response

Introduction:

  • Citation numbers were placed inside the sentence next to its associated fact, avoiding “string citation”.
  • Grammar correction suggestions were attended.
  • Method descriptions were removed from this section.

Materials and methods:

  • Citation numbers were placed inside the sentence next to its associated fact, avoiding “string citation” whenever possible. However, we consider it necessary to cite more than one reference when the sentence describes a method or analysis, because we include a reference for the analysis description and one example of its use in similar contexts. Also, we carefully examined other articles published by Plants and we found that “string citation” is not only permitted, but also, quite common.

Results:

  • Figure 8 and Table 4 were reported in the text before placement.

Discussion

  • Sentences that used “recently” and “nowadays” but cited references from several years ago, were corrected and clarified.
  • Paragraph that stated the relevance of our work was removed from this section and included in the Conclusion, as requested.
  • We explained putative reasons and consequences of our findings whenever possible; however, the aim of this study is a description of the flora of the Revillagigedo Archipelago and arrival of alien plants. We do try to explain similar processes on other island regions of the world. Therefore, we do not hypothesize on potential processes in other regions.

English

  • We conducted an additional review of English grammar and style.

Round 3

Reviewer 2 Report

No issues